# Measuring Students Acceptance and Usability of a Cloud Virtual Desktop Solution for a Programming Course

Francisco J. Rodríguez Lera [1,*,†] , David Fernández González [1] , Francisco Martín Rico [2] , Ángel Manuel Guerrero-Higueras [1] and Miguel Ángel Conde [1]

[1] Department of Mechanical, Computer Science and Aerospace Engineering, University of León, Campus de Vegazana, s/n, 24071 León, Spain; dferng@unileon.es (D.F.G.); am.guerrero@unileon.es (Á.M.G.-H.); mcong@unileon.es (M.Á.C.)
[2] Intelligent Robotics Lab, Rey Juan Carlos University, 28943 Madrid, Spain; francisco.rico@urjc.es
* Correspondence: fjrodl@unileon.es
† Current address: Escuela de Ingenierías Industrial e Informática, Campus de Vegazana, 24071 León, Spain.

**Abstract:** Virtual desktops in cloud scenarios play a significant role in higher education. Nowadays, the idea of moving laboratories to the cloud seems mandatory and it is necessary to maintain students' commitment in this new scenario. This paper aims at two targets, customizing a Virtual Desktop platform for delivering the laboratories of a programming course in a Computer Science Bachelor Degree and empirically apply the technology acceptance model and the system usability scale to a set of students that use it. Results obtained in this paper provide insights about the direct effect between the perceived ease of use, perceived usefulness, and attitude to technology following the technology acceptance model (TAM) as well as a comprehensive analysis of the system usability scale (SUS) of our platform.

**Keywords:** virtual desktop infrastructure; online learning; technology acceptance model; system usability scale

## 1. Introduction

Nowadays, academic experience is growing and adapting to current scenarios of physical and temporal restrictions using synchronous and asynchronous learning mechanisms. To deal with these new scenarios several online learning platforms are available in different cloud shapes for providing teaching and laboratories on demand [1–3]. However, the big number of applications needed by the students and the learning curve required for installing and using the cloud solutions or the software tools, demands some effort from both teacher and student perspectives. In addition, there are extra issues such as those proposed by Shirzard [4] not only related to procuring hardware or software but also allocating spaces or hiring personnel.

There are many online strategies for learning delivering systems in cloud scenarios [5] such as learning content management system (LCMS), LMS (learning management system), massive open online courses (MOOC), or just a video conference system like Zoom [6]. These strategies have to maintain the same goals. From the students' perspective [7], the goal is to be able to receive a course, take exams or send homework. From the teachers' perspective [7], they should be able to provide the content, prepare tests, communicate with students or send feedback. The current market of cloud services supports a range of off-the-shelf approaches to provide effective solutions adapted to the needs of remote teaching and class assistance. One of the most extended solutions is to use virtual desktop infrastructure (VDI) through cloud-based engines [8]. The idea is to provide access to a personal computer environment for the teacher and the student using Internet [9,10] and if this computer could be shared between both simulating a face-to-face laboratory, would be even better.

### 1.1. Problem Statement

Educational organizations are facing new scenarios of attendance at their courses these days. Classic face-to-face and online approaches have included different schemes, mainly when facing laboratory lessons [11]. The teacher is not physically in front of the student's computer, thus it is complicated to assume that the student is going to face the exercises in the same manner as in the laboratory [12].

In addition, there is a new issue associated with creating your laboratory environment: prerequisites such as obtaining specific hardware or installing specialized software are tasks that are loaded in students' working hours. These tasks that previously were solved locally by teachers or lab technicians with the skills, now are fulfilled by students with no knowledge in some cases.

These scenarios open several challenges, not only to the students but also to the teachers that would require spending some time configuring remotely specific machines for each student.

### 1.2. Research Question

The main objective of the research is to shed light on the facilitation of virtual desktop infrastructure, which means a computer in the cloud and its impact on the regular computer science course itinerary.

The research question considered here is:

**RQ:** What are the student's expectations and impact of a Virtual Desktop Infrastructure (VDI) solution designed and adapted for a Computer Science laboratory in the first course.

Different delivery methods for building knowledge are used in the classroom. These days teachers deliver content using infographics, videos, discussions, students pitches, teacher-led sessions, interactive guides, group participation and discussion, hands-on exercises, on-the-floor training, mentor shadowing, or huddles, for mentioning some of them [13,14]. Changing face-to-face models to synchronous mixed online approaches requires a different viewpoint [15]. This is especially relevant when the COVID-19 pandemic arises. Many of the activities carried out face to face must switch to online scenarios which have an important impact in laboratory work [16–18]. Given our intention to use virtual desktop infrastructure in our courses, it is necessary to evaluate usability and acceptability first.

This study poses four main hypotheses that should be validated attending our usability and acceptability premises:

**Hypothesis 1 (H1).** *The VDI perceived usability by the students is considered correct using a predefined scale.*

**Hypothesis 2 (H2).** *There is a significant relationship between Perceived Usefulness of student and Attitude Towards Technology.*

**Hypothesis 3 (H3).** *There is a significant relationship between the Perceived Usefulness of students and the Perceived Ease of Use.*

**Hypothesis 4 (H4).** *There is a significant relationship between Perceived Ease of Use and Attitude Towards Technology.*

The H1 hypotheses will be validated with the system usability scale [19]. Hypotheses H2, H3, and H4 are handled and validated with the technology acceptance model [20].

Collaterally, it is performed a brief Sustainability analysis of the approach from an economic perspective. It provides an alternative vision when providing virtual desktops to students. Highlight that SUFFER [21] was completely deployed in a private server located in group premises.

## 2. Materials and Methods

### 2.1. Literature Review

This project explores the idea of a Cloud computing-based desktop virtualization such as a virtual desktop infrastructure (VDI) or remote desktop service (RDS) [22]. The idea of this technology is to enable access to a personal computer environment regardless of the connecting device, at any time or place as long as the user has Internet available.

To provide a suitable environment to students at the beginning is one of the purposes of delivering our laboratory classes in higher education. Attaran [23] research addresses the different phases of a cloud service adoption, proposes strategies, and explores the main factors that may contribute to its success in the world of education. However, it is well known that technology acceptance is not a straightforward process given cultural and societal restrictions [24,25]

The interest in establishing an online environment ready for teachers and students has also grown with the COVID-19 scenario. Researchers from different countries [26] have been working on this topic analyzing the impact of Cloud computing on teaching [27].

Researchers such as Hossain [28] or Erskine and Füstös [29] have been performing the cost-effective analysis of deploying this kind of tools. Thus, this study also explores a brief sustainability overview from the economical point of view.

This study aims to understand the student perspective about these platforms, not only from the TAM perspective as in [30,31], but also from a system usability Scale as Pal et al. did in [32] for comparing different online tools such as Microsoft Teams. Notwithstanding, this study performs the analysis of each value independently rather than performing an extension of TAM as in [33]. This is because our Virtual Tool, SUFFER [34], is designed and developed by some of the authors at the same time is being used, so we have focused on the technical part and trying to take advantage of the SUS scale, to enhance its performance and continuing using it next years.

### 2.2. SUFFER Platform

This study is based on the SimUlation Framework for Education in Robotics (SUFFER). It was designed and developed to promote the robotic field remotely. The idea is to simplify the deployment process and to facilitate a system where the teachers can simulate robots and supervise students. Figure 1 illustrate the design and the interaction with students and Teachers. The initiative proposes to deploy virtual desktops, disposing of a robot middleware such as ROS (Robot Operating System) or YARP (Yet Another Robot Platform), simulation tools, and a set of datasets obtained by real sensors. The aim is to focus on the course and the content involved, avoiding some of the tedious tasks of system configuration to the student.

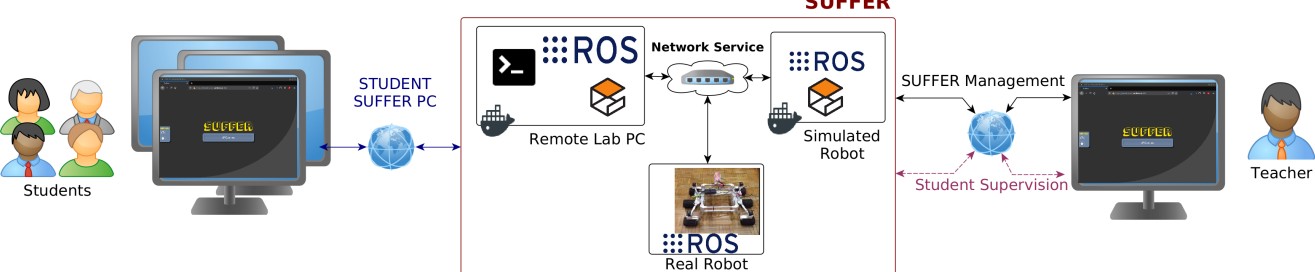

**Figure 1.** SUFFER architecture.

This first objective was to design a solution easy to adapt generalized to be used with any other middleware, data, and application, which becomes SUFFER in a very flexible cloud platform to define remote labs. The main advantages of SUFFER compared with usual commercial remote Labs are the benefits of adding customization levels managing the information on local premises. This feature allows us to offer Container as a Service

(CaaS), Platform as a service (PaaS) Software as a Service (SaaS) perspective. This means to offer machines with classroom technologies.

SUFFER allows student monitoring and guiding during the practice on-demand, it implies providing support from teachers in real-time on our system, rather than the classic message interchange approach. Teachers can access any machine and correct the problems the students have or lead them in solving the problems. The work proposed in [21] provides an example.

### 2.3. Teaching Approach

The study is based on a computer science course. It aims to provide a broad and robust understanding of computer science and programming languages to first-year students. Particularly, during the laboratories, the student reviews basic object-oriented concepts such as abstraction, algorithm, data structures, or encapsulation. These laboratories are focused on practical exercises where the students solve different problems programming efficiently. The physical laboratory environment is supported on a set of 20 computer desktops running Ubuntu 20.04. They provide an editor that the student chooses between Gedit and Visual Studio Code. The main programming language used during the course is Java in version 11.

SUFFER is used in each Laboratory session by teachers and students. It simulates the physical environment in the cloud. The teacher shares the screen with a traditional meeting application (Google Meet). Teachers emphasize reaching a given practical target, thus, the students have to practically test most of the exercises proposed during the laboratory. In case that a student finds an issue, the teacher connects to the student's desktop and helps to solve the issue.

### 2.4. Cloud Platform Features

The hardware deployed for this experiment is partially shared with other projects of the group. The specifications of the AMD EPYC$^{\text{TM}}$ is a machine with 16 CPU Cores and a total of 32 Threads. The server has a 3.0 GHz Processor with an L3 Cache of 128 MB deploying 264 GB of RAM.

### 2.5. Virtual Desktop Description

We have deployed 148 virtual desktops on the server. This means a total of 338.2 GB measured with the Linux command *du*. The average space used by each platform is 2.3 GB, with a mode value of 2.1 and a median of 2.3 GB. The min value is 2.1 GB and the max value is 3.4 GB. However, there are a few cases that overpass this space because if an application fails, it is generated a file containing an image of the process's memory after the application was finished abruptly.

### 2.6. Demography

This experiment evaluates the adaptation of SUFFER to a Programming I course. To smooth the name SUFFER for its English meaning, we decided to change its name following a naming convention used in the group that establishes to choose a name from the cast of a Cartoon TV Series, in this case, we choose CINDY by Yogi Bear series. The course is taken by 148 students and is taught by three teachers. It is delivered in the first course of Bachelor Degree in Computer Science Engineer, located in the Industrial, Computer Science and Aerospace School of Engineering of University of León, Spain. The numbers show 23 women and 119 men. The class is composed of 105 first-year students, where the typical age for entering university in Spain is 18 years old, and the others are students that failed the course previous years and they need to study again.

SUFFER was developed to ease the interaction between teacher-student during COVID restrictions. Its configuration was oriented to minimize or remove any personal information stored by the students. The students were informed of SUFFER's purpose and each study participant had the right to withdrawal from using SUFFER at any time or not to implement

our questionnaires. Using the tool was not necessary to pass the course and we did not offer any kind of reward for its use. The students have a guide for installing the environment in their computers, this also solves the issues with those without an Internet connection at home. The questionnaires planned for this study were defined paying attention to not personalizing any of the responses of the respondents. Besides, during the semester we asked them about their subjective opinion about current performance and future enhancements.

### 2.7. Questionnaires

The questionnaire proposed in this study was available to all students in the course (for those that do or do not use SUFFER). For those that used it, we perform a questionnaire that contains 30 questions. First, four questions dealing with the demography. Then, ten questions are associated with the technology acceptance model (TAM) defined by Davis [20]. Ten questions associated with the system usability scale (SUS) [19] and finally a set of extra questions covering qualitative and quantitative issues associated with specific circumstances of the solution and perceived ease of use.

The methodology followed to analyze these questionnaires is a mixed approach [35] that considers both quantitative results such as those obtained from SUS and TAM and qualitative results following a matrix approach by answer proximity [36].

#### 2.7.1. TAM

In TAM, Davis focuses his work on perceived usefulness, perceived ease of use, and user acceptance of information technology. The main goal of Davis was to check two factors according to the individual's intention when using new technology. These two factors are the perceived ease of use and the perceived usefulness. Some works remember its weaknesses [37]; however, previous experiences of authors consider this questionnaire a good approach for getting a quick overview of adults' intention to use new technology.

The technology acceptance model (TAM) is one of the classic mechanisms for modeling how an individual utilizes new technologies. The model proposes a set of factors that drives to know behavioral intention to use technology from an end user's perspective (Figure 2).

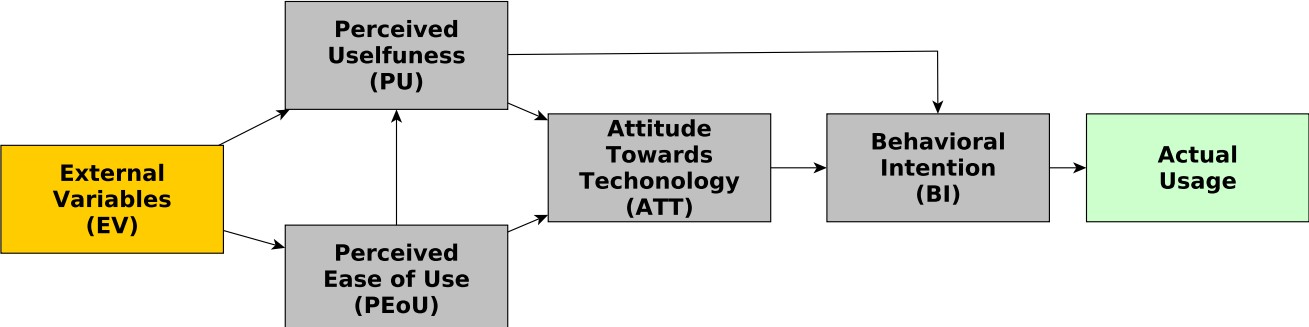

**Figure 2.** TAM Model diagram based on [20].

The core of the TAM questionnaire used in this study contains a set of questions associated with three different factors: perceived ease of use, perceived usefulness, and attitudes toward technology. Perceived usefulness (PU) defines the degree to which an individual believes that the system would enhance an activity performance. Perceived ease of use (PEoU) defines the degree to which an individual believes that a system will be free of effort These two motivations are considered determinants to shed some light on the behavioral intention to use a system. PU is considered a fundamental factor for knowing the technology acceptance rate and PEoU predicts the user's acceptance or rejection rate of technology.

TAM model has to be valid to assess the behavioral intention of use SUFFER by the first-year students during our course of programming taught in the first year of Bachelor

Degree in Computer Science. Table 1 overviews the three factors and covers the constructs used in this study.

**Table 1.** TAM Questionnaire proposed in this study (English version).

| Variables | Item-ID | Item |
|---|---|---|
| Perceived | TAM-Q1 | CINDY would improve my learning performance |
| Usefulness | TAM-Q2 | CINDY would improve my academic productivity |
| (PU) | TAM-Q3 | CINDY could make it easier to study course content |
| Perceived | TAM-Q4 | I find CINDY easy to use |
| ease of use | TAM-Q5 | Learning how to use CINDY is easy for me |
| (PEOU) | TAM-Q6 | It is easy to become skillful at using CINDY |
| | TAM-Q7 | I have the necessary skills for using CINDY |
| Attitude | TAM-Q8 | Programming through CINDY is a good idea |
| Towards | TAM-Q9 | I am positive toward CINDY |
| Tech.(ATT) | TAM-Q10 | I intend to be a heavy user of CINDY |

### 2.7.2. SUS

System Usability Scale was created in 1996 by John Brook [19]. Even though the title is a bit tricky, it is a reliable and popular questionnaire accepted by most usability professionals. It consists of 10 items that provide a usability score in the range of 0–100 using a simple scoring system. Then this score fits in a scale that allows to classify the ease of use of something tested. The English version of our questionnaire is presented in Table 2.

**Table 2.** SUS Questionnaire proposed in this study (English version).

| Item-ID | Item |
|---|---|
| SUS-Q1 | I think that I would like to use CINDY frequently. |
| SUS-Q2 | I found CINDY unnecessarily complex. |
| SUS-Q3 | I thought CINDY was easy to use. |
| SUS-Q4 | I think that I would need the support of a technical person to be able to use CINDY. |
| SUS-Q5 | I found the various functions in CINDY were well integrated. |
| SUS-Q6 | I thought there was too much inconsistency in CINDY. |
| SUS-Q7 | I would imagine that most people would learn to use CINDY very quickly. |
| SUS-Q8 | I found CINDY very cumbersome to use. |
| SUS-Q9 | I felt very confident using CINDY. |
| SUS-Q10 | I needed to learn a lot of things before I could get going with CINDY. |

### 2.7.3. Extra Questions

Finally, the questionnaire covers three extra items: (1) network quality at home and in university. (2) perception of the tool after a massive update that took place in the middle of the semester. (3) qualitative point of view: (a) what is your opinion about the tool, (b) could you enumerate the advantages, and (c) could you enumerate the disadvantages. It can be tracked here: https://forms.gle/xGriQdo4VeCqwKT6A (accesed on 7 July 2021).

The quantitative questions present the items to participants using a five-point scale. They are numbered from 1 to 5 anchored with "Strongly disagree" or "Strongly agree" in each item. The use of a center scale numbered as 3 could be considered as a neutral or failing item, attending the question.

## 3. Results

A total of 148 virtual desktops with our Programming course were established for the course. A total of 139 students made an assignment attempt (97.98%), this value was obtained just by searching for the name of the zip files provided during the course. It analyzed the total space consumed in our Volume by students without paying attention to the content on it stored. The total space used by our machines in these three months is presented in Table 3. These figures gave us an idea about the hardware requirements needed for escalating this project. The average column presents the typical container after the student starts and does one or multiple practices. Exception (1) and (2) presents the two exceptional cases. The former were those containers where the students were downloading non-related files, and the latter the cases where the student did not open the machine. In the first case was necessary to notify the student for using the platform just for educational purposes. The second case gave us an idea of how many students were not interested at all in the platform.

**Table 3.** Cindy Space Consumption.

|  | Average | Exception (1) | Exception (2) |
|---|---|---|---|
| Mean | 180.73 MB | 4.33 GB | 149 kB |
| Mode | 5.27 MB | - | 140 kB |
| Median | 149 MB | 1.66 GB | 149 kB |
| Standard Deviation | 167.63 MB | 4.55 GB | 0 kB |
| Minimum | 4.83 MB | 1.07 | 149 kB |
| Maximum | 679 MB | 11.1 | 149 kB |
| Sum | 25,121.81 MB | 25.96 GB | 447 kB |
| Count | 139 | 6 | 3 |

Of the total of 148 students, 42 (28.38%) answered the questionnaire. Of these respondents 35 were males, 6 were women and 1 decided not to answer. A total of 41 of these students were in their first year and 1 in the second year. The age of the students was bounded in blocks where 78.58% of the students were 17–18 years old.

### 3.1. TAM

Figure 3 presents graphically the results of the TAM questionnaire prepared for this study. As presented above, it was used Likert scale from 1 to 5.

Firstly, it measured the reliability test of the questionnaire using Cronbach's alpha coefficient $C\alpha$, the composite reliability value $CR$, and the Average Variance Extracted (AVE). To this end, it was used JASP and the reduction factor from SPSS (for CR and AVE values). The results of the tests demonstrated scores for all the items over the classic criterion (0.5) as shown in Table 4. The Cronbach's alpha coefficient for 10 items is 0.911 and the independent factors were also validated, therefore the constructs in this questionnaire show sufficient reliability. Individually, PU and PEoU present reliable values, and ATT a slightly low value (0.68); however, given the overall results and values used by the Science Education community [38] which consider reasonable this value, we will continue these constructs.

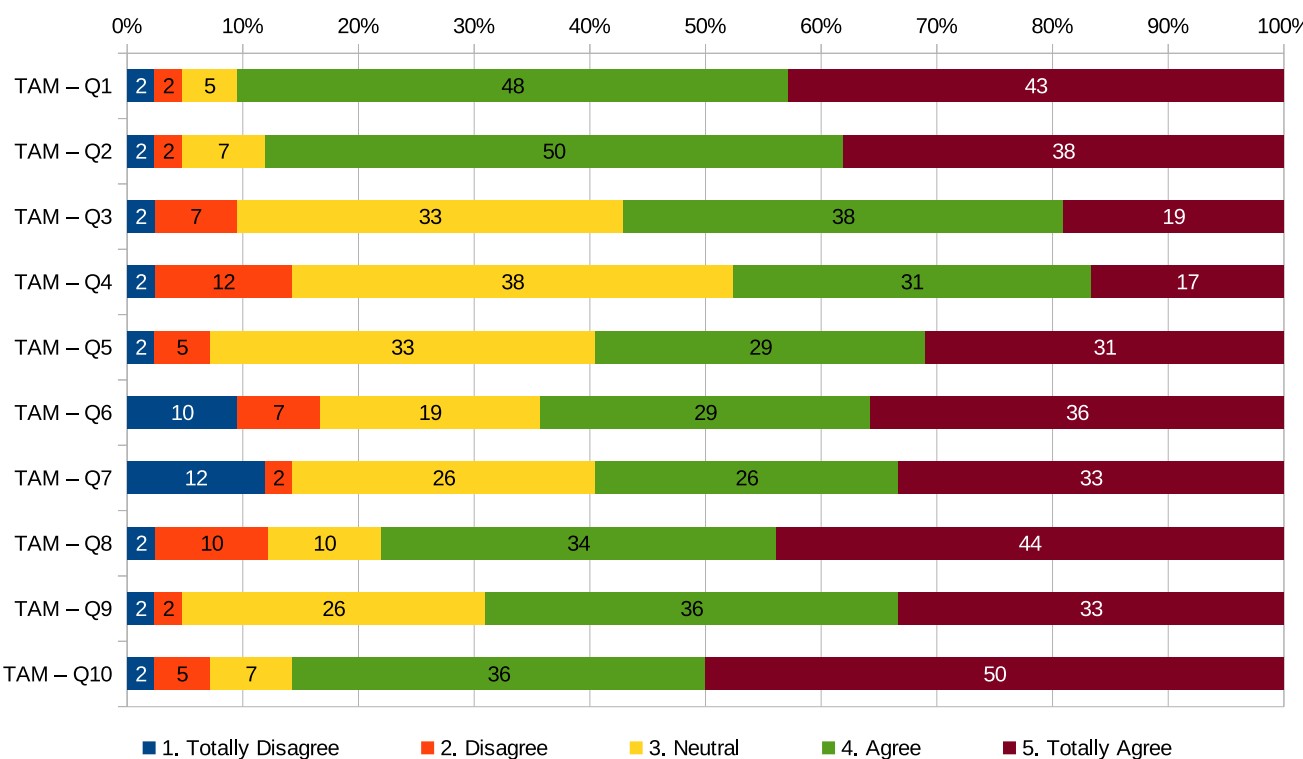

**Figure 3.** TAM questionnaire answers in percentage.

**Table 4.** Frequentist Scale Reliability Statistics. PU: Perceived Usability, PEoU: Perceived Ease of Use, ATT: Attitude Towards Technology, AIC: Average Inter-item Correlation.

| Factor | Estimate | Cronbach's $\alpha$ | AIC | Mean | sd | CR | AVE |
|--------|----------|---------------------|-----|------|-----|-----|-----|
| Overall | Point estimate | 0.911 | 0.513 | 3.883 | 0.287 | | |
| PU | Point estimate | 0.890 | 0.746 | 3.650 | 0.185 | 0.860 | 0.672 |
| PEoU | Point estimate | 0.889 | 0.674 | 4.067 | 0.307 | 0.893 | 0.678 |
| ATT | Point estimate | 0.681 | 0.420 | 3.870 | 0.214 | 0.832 | 0.633 |

This study proposes to perform the structural model fit testing attending the significance of the coefficients. Using the AMOS tool, it is obtained the estimated value of the coefficient, the standard error (SE), critical ratio value (CR), the probability (P) (showing *** means <0.001) for each coefficient. The relationship between factors is significant when it is obtained at the 95% confidence level. Thus, the critical ratio value (CR) is $\geq$1.96, and then the correlation is significant on the level $\leq$0.05 [39]. Table 5 presents our results where all the CRs are bigger than 1.96 and $p$ values are under 0.05, therefore, the hypotheses formulated are accepted.

The perceived usefulness presents a significant effect on perceived ease of use. Attitude towards technology and perceived usefulness has a strong effect on each other and perceived ease of use and attitude towards technology has also a strong effect. Table 5 presents the results of confirmatory factor analysis developed in AMOS. These values validate that our constructs variables can be reduced into three factors.

**Table 5.** Covariances between factors.

| | Estimate | S.E | CR | $p$ |
|--------|----------|-----|-----|-----|
| PU $\leftrightarrow$ PEoU | 0.382 | 0.131 | 2.914 | 0.004 |
| ATT $\leftrightarrow$ PU | 0.710 | 0.215 | 3.308 | 0.001 |
| ATT $\leftrightarrow$ PEoU | 0.505 | 0.173 | 2.922 | 0.003 |

These three factors—perceived usefulness, perceived ease of use, and attitude towards technology—have exhibited a stronger effect on user's attitudes and behavioral intention to use CINDY. This implies that perceived usefulness and perceived ease of use are two crucial factors to guarantee SUFFER adoption in the course.

SUS

The system usability scale scores are computed for the 42 students who completed the questionnaire. Figure 4 shows the answers obtained after using CINDY in percentage.

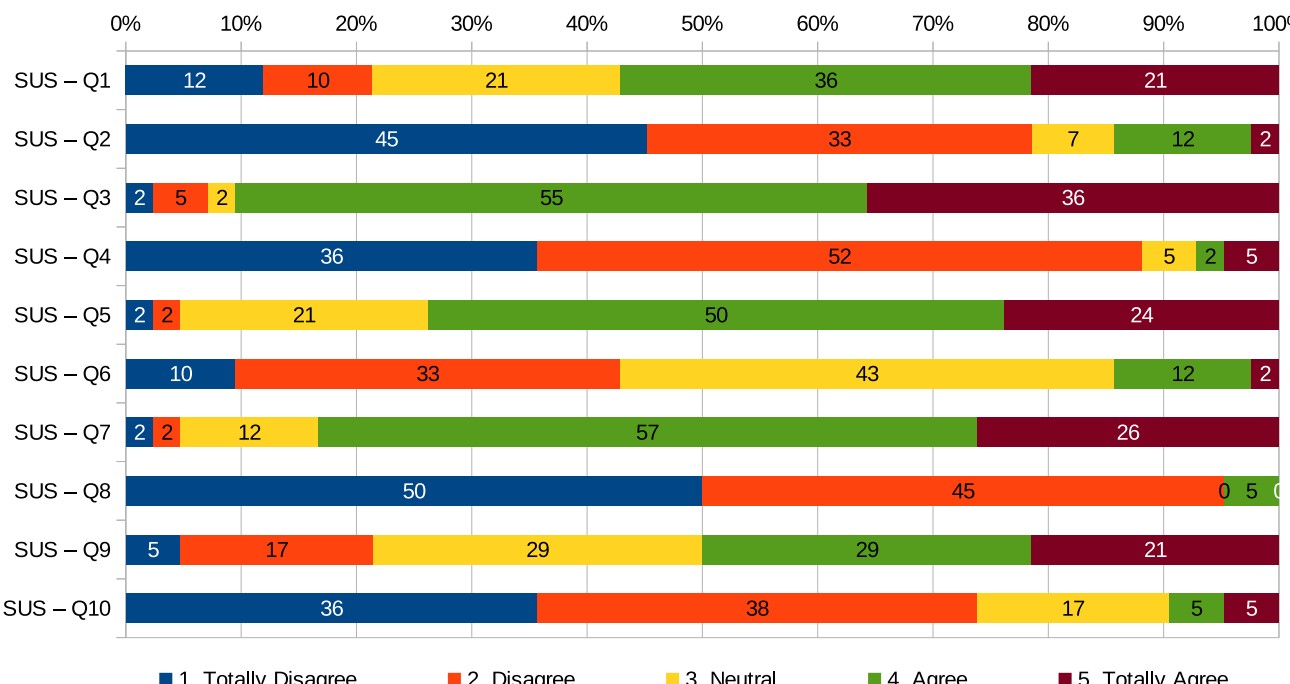

**Figure 4.** SUS results.

Four SUS scores are calculated against the estimated usage metrics for all students: total value and grouped by gender, age, and Internet connection. The total item displays the overall score of the 42 students. Gender fields were presented as male, female, or preferred not to say. Age fields were grouped also for minimizing the information of each student. Besides, because the proposed system is based on cloud services, it was asked if the respondent has an Internet connection at home.

The calculation is determined by each item's score contribution ranging from zero to four following the classic approach. It is necessary to count those positively-worded items enumerated by 1, 3, 5, 7, and 9. The score contribution of these items is defined by the scale value minus 1. Those items considered negatively worded (2, 4, 6, 8, and 10) define the score subtracting 5 to the scale position. Finally, the overall SUS score is obtained by multiplying the sum of the item score contribution by 2.5 and as a result, we get the scores to range from 0 to 100. This scale reflects higher satisfaction from the user with higher values. General SUS interpretation shows [40] the next adjectives for the scale: Excellent [>80.3]; Good [68–80.3], Okay [68], Poor [51–68], and Awful [<51]. Updated approaches recognized more adjectives [41] Best Imaginable [>90.9], Excelent [90.9–85.5], Good [85.5–71.4], Okay [71.4–50.9], Poor [50.9–35.7], and Awful [35.7–20.3], Worst Imaginable [<20.3]. Figure 5 illustrates graphically the results of our questionnaire associated with each researcher study.

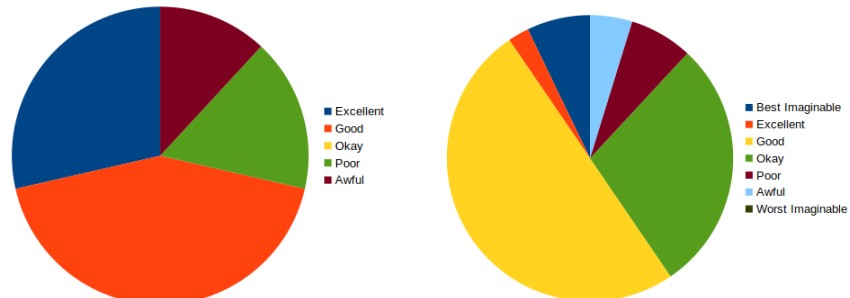

**Figure 5.** SUS results: (**left**) using classic scale [40], (**right**) updated approach [41].

Under these results, it is obtained a 43% okay and 29% Excellent with 27% in Poor and Awful. Under Bangor approach 7% Best imaginable, 2% excellent, 50% good and 28% Okay. Poor, awful, and worst imaginable bound almost 12%. Table 6 overviews the descriptive statistics assiciated to SUS questionnaire.

**Table 6.** System Usability Scale: Total overview.

|                 | Total |
| --------------- | ----- |
| Valid           | 42    |
| Mean            | 72.26 |
| Median          | 72.50 |
| Mode            | 70.00 |
| Std. Deviation  | 14.33 |
| Minimum         | 32.50 |
| Maximum         | 95.00 |
| 25th percentile | 67.50 |
| 50th percentile | 72.50 |
| 75th percentile | 82.50 |

Finally, it is explored the SUS attending the age, gender, and Internet Connection availability. Figure 6 shows the boxplot for the different age groups present a *Good* result for all ages except for 23–25 and 25, the former within the group of okay and the latter within the group of Excellent. Center of Figure 6 presents the boxplot by Gender and the right diagram by the Internet connection. The reported SUS results by sex find slightly better perception than Male or not said cases; however, classifications bounded them in the same window *Good*. It analyzes the answers delivered by those with no Internet at home. It presents a lower ratio (under%1) than those with the Internet at home. However, it is important to say that this small percent is key when using Bangor scale [41] because it means that it goes from *Good* to *Okay*.

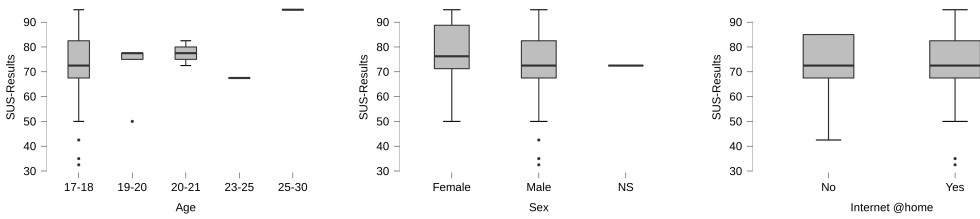

**Figure 6.** SUS boxplots: (**left**) by age, (**middle**) by gender, (**right**) by internet connection.

## 4. Discussion

This research conducted a study to empirically answer the RQ presented at the beginning of this paper. Checking the production environment this study considered that those machines with an average volume consumption of 2.1 GB have been used and those with higher space consumed. Under these values, it was measured a total of 39 machines, which

means that corresponds with a set of students that did not try to follow the course with our solution.

Initially, we analyzed the technology acceptance model by assessing the constructs of the model in the context of the Computer Science course. The results validated the three hypotheses H2, H3 and H4.

Likewise, the hypotheses H1 was validated. This study presented not only the general score, which fits in the SUS *Good* objective window. We explored a completed a fine-grain analysis of the data for validating SUS alternatives. The results were always quite positive except for those with no internet connection at home. However, it is not possible to grow up under this hypothesis because some of the sample groups are not significant (for instance, there were just two individuals between 23–30 years old).

Finally, it is evaluated the qualitative answers delivered in the three open questions proposed in the questionnaires. Regarding the qualitative analysis, we include a sample of 11 of the student's answers from the 41 gathered. These answers are shown in Table 7 and they are distributed by proximity criteria as suggested by Miles and Hubberman [36].

**Table 7.** Students answers to qualitative questions.

| ID | EXT-Q1 Opinion about Cindy | EXT-Q2 Advantages | EXT-Q3 Disadvantages |
|----|---------------------------|-------------------|----------------------|
| 1 | It is a very useful tool | All tools installed | Screen resolution |
| 2 | It is very easy to use | Easy Setup | Slow connection |
| 3 | It helps a lot at the beginning. | Avoid OS Installation | Network dependence |
| 4 | It lacks some functions | Remote Supervision | Lag issues |
| 5 | Overall Very good | Intuitive | Screen quality |
| 6 | Useful for Linux newbies | Easy to Access | It hangs suddenly |
| 7 | Good tool | Complete workspace | Keyboard mapping |
| 8 | it works surprisingly well. | Doesn't take laptop resources | Missing tools from other courses |
| 9 | Great idea | Teacher supervision | Slows down in delivery days |
| 10 | It is an interesting tool | Easy to start programming | I feel observed |
| 11 | Double Check System | Jump easily between computers | Reboot at 4:00 am |

The first question (EXT-Q1) requested to show their opinion about CINDY with their own words. The answers present positive (very good, great idea, quite useful at the beginning) and some negative comments (lacks some functions). The former was focused on its characteristics of "Plug-and-Play", mainly for newcomers, as well as to have a clone machine from the laboratory to double-check the results when the student is not sure about the root of an issue, their computer, or their solution. The latter was focused on some functions that are missing on CINDY such as other course tools or some social media blocking in the browser for avoiding student distractions.

Question two and three (EXT-Q2 and EXT-Q3) were focused explicitly on the advantages and disadvantages of CINDY, to avoid random comments in question 1. Thus, the set of answers were categorized into categories, either objectively or subjectively measured:

1. Usability I: Some students valued positively the utility of remote attendance by teachers synchronously.
2. Usability II: the efficiency of arriving at the class and home and finding everything installed and the last exercise that they did in the class.
3. Usability III: flexibility, some of the students proposed a mechanism for working in the cloud and locally indistinctly for avoiding the network problems.
4. Usability IV: robustness against network issues. They highlighted the need for a network for being able to continue their practices, and some of them suffer issues from their homes.
5. Performance expectancy: some of the students are not happy with its performance, given parameters such as lag that creates delays between its interaction and the results.
6. Robustness: two students complain about the maintenance reboot programmed every day at 4:00 a.m. This behavior was necessary to reset some of the most expensive computational and memory applications running in the system.

7. User Experience I: for some of the students with different ratio monitors experience issues. It was established by the default configuration of 1920 × 1080. This means re-scaling for some of the users, generating bad perception by students.

8. User Experience II: Some students raised doubts about their privacy. They said that the teacher would "spy" their sessions.

These were the main factors that respondents answered in our questionnaire. Most of these answers revolve around a single problem, connectivity issues. Thus, no network, no machine; therefore, they feel that the solution is too dependent on this parameter. Some students found in the application a good approach to start the first year, and they suggested convincing other teachers integrate their tools in the time that they proceed with the installation of the Operating System and tools required for the day today. In addition, the robustness category about the maintenance process was surprising because it was not expected of any user at that time, and the duration of the process was under five minutes. The issues associated with screen quality are mainly focused on the configuration of the desktop and the Xorg server on each system. However, to minimize vulnerabilities in our infrastructure, the students do not have permission and the ratio was calculated automatically and some students experience bad resolutions in their computers.

*Sustainability*

Sustainability concepts are becoming critical in the current decision-making processes of higher education. Thus, it would be great to perform a sustainability overview from the economical perspective to measure the cost of introducing new technologies in the current toolbox of the classroom. To this end, we are going to calculate CINDY deployment cost using a service offered by a third-party company.

The machines are running every day and the students have full access to CINDY 24/7. The production environment started at the beginning of the first semester in September and we finished the measurement at the beginning of December, which was the end of regular laboratories for focusing on final projects. At this time the machine was rebooted 2 times. On the 16th of October, for adding a massive update of workspaces, and the 4th of December, the machine suffered a fail in a software update in a third-party library and was necessary to reboot it.

The first step is to perform an analysis of the resource allocation. This will provide a footprint supported on four elements: Server CPU Performance, RAM, Operative System and User dedicated volume. Given that our laboratories are regular PCs with 4 GB of RAM on Ubuntu, we decided to estimate something similar. Secondly, we identified the software footprint of those applications installed in the classroom: Java and Visual Studio Code. Java space footprint is less than 150 MB (https://www.oracle.com/java/technologies/javase/windows-diskspace.html (accessed on 7 July 2021)). The RAM required for our programming scenario is minimal (always under 10 MB). Visual Studio Code requires less than 200 MB of volumefootprint (https://code.visualstudio.com/docs/supporting/requirements (accessed on 7 July 2021)). However, there is a problem with this tool, that it requires at least 1 GB of RAM. Thus, these figures would be satisfied by 4 options as presented in Table 8. There are four commercial options (two by the company) that fit with our approach. The main differences are the RAM requirements and the number of vCPU.

**Table 8.** Cost associated with a desktop unit.

|  | vCPU | RAM (GB) | Disk (GB) | OS | Price (Month) | Price (Hour) |
|---|---|---|---|---|---|---|
| Option 1. Microsoft A1V2 | 1 | 2 | 10 | Ubuntu | 26.51 * | 0.36/h |
| Option 1. AWS | 1 | 2 | 10 | Ubuntu | 25.00 | 0.20/h ** |
| Option 2. Microsoft B2S | 2 | 4 | 8 | Ubuntu | 21.96 * | 0.30/h |
| Option 2. AWS | 2 | 4 | 10 | Ubuntu | 33.00 | 0.30/h ** |
| Personal Computer | Celeron N4020 | 4 | 32 | Ubuntu | 299 |  |

* Considered 730 h; ** Fixed: +$9 month.

Table 8 presents the estimation cost per month of different commercial approaches. Rounding, it presents a semester of 4 months and 100 students for about 10,400 EUR. This means that 100 students could perform the course at the same time. Supplementary to these options, it is necessary to calculate the device employed to access the server, in this case, it is proposed a low-cost computer. Again, rounding the cost implies spending 30,000 EUR for a full class of 100 students or just 9000 EUR for preparing a laboratory of 30 places. To summarize, a center that requires teaching using this kind of method covering all the expenses has to think in a budget of 40,000 EUR.

Although the second step associated with the sustainability topic is to face a possible reduction of the $CO_2$ footprint it requires its research. As said, using a virtual desktop requires, on the one hand, the server or servers that guarantee the high-performance CPU, disk availability, etc. On the other hand, it requires a device by a student, the local hardware such as a tablet, laptop, or desktop computer. Thus, it is necessary to perform the $CO_2$ analysis with a deep analysis by hardware platform as other researchers do [42,43].

## 5. Conclusions

Returning to the subject, this study evaluated the use of our VDI solution called SUFFER using two different mechanisms for knowing more about its use for a computer Science course. The idea was to provide a set of answers to the proposed research question: *What are the student's expectations and impact of a Virtual Desktop Infrastructure solution designed and adapted for a Computer Science laboratory in the first course.*

In conclusion, these kinds of infrastructures are well received by students and teachers given the ease with which a laboratory can be deployed and this conclusion was extracted using some quantitative (TAM and SUS) and qualitative (user opinions) analysis.

The advanced technology acceptance model (TAM) was applied to assess the behavioral intention to use SUFFER during the course. Perceived usefulness of SUFFER was found to be the strongest determinant factor for their attitude followed by Perceived ease of use that also has a strong significant effect on their attitude. The results presented just a significant relation between the perceived usefulness and perceived ease of use.

On the other hand, it was analyzed the System Usability Scale, obtained a score of 72.26 which is considered *Okay* under the umbrella of original John Brooke or Bangor studies.

After this experience, we believe that the tool is okay for use with first-year students of Computer Science and it is expected to enhance students' results if adapted correctly. This study found some issues and positive elements of impact on the students' satisfaction. However, it is clear that the drawbacks of this kind of solution are its high network dependence, thus it has to be clear the network conditions of the place where this kind of solution will be used.

Furthermore, it is necessary to analyze students' perception of privacy in this tool of remote teaching. Some of them exposed that they would feel observed during the laboratories, even the teacher supervision is on-demand, and the teachers were only connected to those machines where a student required help.

**Author Contributions:** Conceptualization, F.J.R.L. and M.Á.C.; Formal analysis, F.J.R.L.; Investigation, F.J.R.L.; Methodology, F.J.R.L. and M.Á.C.; Software, F.J.R.L. and D.F.G.; Supervision, M.Á.C.; Validation, F.M.R. and Á.M.G.-H.; Writing—original draft, F.J.R.L.; Writing—review and editing, F.J.R.L., D.F.G., F.M.R., Á.M.G.-H. and M.Á.C. All authors have read and agreed to the published version of the manuscript.

**Funding:** This work has been partially funded by the Kingdom of Spain under grant RTI2018-100683-B-I00 and Erasmus+ KA201 Strategic Partnership Project, Grant/Award Number: 2018-1-ES01-KA201-05093.

**Institutional Review Board Statement:** Ethical review and approval were waived for this study, due to two reasons: (1) everything was set for minimizing personal information of the individual. (2) the total number of answers is under the total number of participants.

**Informed Consent Statement:** Individual written consent was waived due to participate in the questionnaire was optional. The individuals known about this study and they could choose if do it or not.

**Data Availability Statement:** Provided on-demand.

**Conflicts of Interest:** The authors declare no conflict of interest.

## Abbreviations

The following abbreviations are used in this manuscript:

| | |
|---|---|
| ROS | Robot Operating System |
| SUFFER | SimUlation Framework for Education in Robotics |
| SUS | System Usability Scale |
| TAM | Technology Acceptance Model |
| VDI | Virtual Desktop Infraestructure |
| YARP | Yet Another Robot Platform |

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
