# Peer review of "Measuring Students Acceptance and Usability of a Cloud Virtual Desktop Solution for a Programming Course"

_applsci, doi:10.3390/app11157157_

Round 1

Reviewer 1 Report

The authors have developed/constructed a platform (called SUFFER/ CINDY) and they tested the users’ satisfaction touching usability and acceptability.

Three types of questions are: TAM-questions, SUS-questions and Extra questions.

I tried to reach the questionnaire through the link in the paper but, unfortunately, I could not.

My remarks: Table 1 entitled „TAM Questionnaire proposed in this study” contains 10 questions. Figure 2 contains 12 questions. Why? First two questions in Figure 2 belong to PEoU, are not they? (Remark: H1,…, H10 in Table 1 obviously differ from H1, H2,..in 1.2, they should be denoted differently).

Cronbach alfa of ATT is a little bit low (under 0.7).

H1 can be considered true concerning TAM. What about SUS?

H2 is investigated in case of SUS only. It is not supported, due the small number of females. I think that the means of males and females do not differ significantly even in this case and, also in case of TAM. Therefore, H2 should be deleted. Age is similar.

I do not know how many teachers use CINDY. In the future, it would be interesting the evaluation grouped by teacher-student if it is possible – there would be enough data.

H3, H4, H5 are right, there are significant correlations between the mentioned types of questions. But: what about the correlation between SUS and TAM?

The authors may investigate the differences between the expected points of PU, PEoU, ATT statistically (for example by t tests).

The explanation (phrasing) in line 257-259 is a little be strange. Confidence level 0.95 means 0.05 significance level. In this case the critical value is 1.96. If CR>1.96, then the correlation is significant on the level 0.05.

The extra questions are very useful, they can really help the improvement of the system.

Summary: statistical evaluation of user satisfaction based on usually applied questionnaires.

The results can help to improve the platform and can support the spreading of the platform.

There are lots of misspellings, the serial number of tables are sometimes wrong (Table 1, Table 2 are twice), the name of Füstös (Fustos) in the References is replaced by its first name (Janos), the third sentence in 3.1.1. has not been finished, inconsistent usages of capital letters very often in References, but also in the sections.

The title is: Understanding Students’ perception of a Cloud Virtual Desktop Solution: TAM and SUS empirical study. Without any doubt concerning the positive properties of the platform, do the author understand the students’ perception? I think that the title is too general.

Summary: the statistical investigation is useful for the developers and for the future users if the platform had spread. I cannot judge whether there is chance for spreading or not.

The paper can be published, but, due to the actual state detailed above, the paper needs revision before publishing.

Reviewer 2 Report

The manuscript touches on the current problem of delivering classes in a virtual manner.

The paper requires a thorough refinement of writing style, presentation of results, and references.

Detailed comments:

line 31 - should be "Infrastructure",
line 70 - it is not clear the meaning of the sentence. Please correct.
line 82 - should be "It provides",
reference [26] missing year of publication.
line 98 - try to use various words instead of "Different researchers in different",
line 101 - for [29] you used last names of authors but in the references section, you wrote first names. You should correct the references section.
lines 107-110 - try to write clearly. One sentence in four lines makes it difficult to understand.
line 111 - it is worth considering include in the manuscript the drawing of SUFFER platform architecture.
line 126 - missing dot.
line 141 - you have used Java language in version 11. JDK is a bundle of tools with JRE. Please be careful.
line 151 - I assume that you mean that the server is equipped with a processor of 3.0 GHz.
line 184 - should be "Then, ten questions"
you have Table 1 two times on page 6 
line 227 - I could not access the enclosed link,
line 238 - should be "Table"
Table 1 after line 239 - there is no clear meaning of "Exceptional 1" and "Exceptional 3"
Figure 2 - please use labels for questions. It will make the figure more readable. The same remark to Figure 3.
line 266 - it appears to me that you meant "into two factors",

lines 267-269 - please refine that sentence,

lines 275-277 - please use various words instead of 4 times "grouped",

Please do check carefully the rest of the text of the manuscript.

Try to avoid citing in conclusions.

lines 445, 451, 458, 465 - need your attention.

References - should be written according to required style.

Reviewer 3 Report

The authors presented an interesting and timely research: they used 148 virtual desktops during a Programming I. course for 148 students. Out of them, 42 answered questionnaires which served as the basis for evaluation.

The methods are clearly presented. However, the presentation of the results could and should be improved. Perhaps the authors should present them in more detail. Also, in line 274, the following sentence can be found: "As explained, it was"... It was what?

Also, if a user did not have internet at home, how did they access the cloud? Did they need to access it before the evaluations?

Maybe boxplots should be used instead of Tables 4 and 5.

I believe that the Item IDs in Table 1. should not use the letter H. It could confuse the readers, because H is used for hypothesis.

Also, the linked Google Form requires permission, to access. Do not include it in the article if it is inaccessible.

English language in the article requires moderate changes (at least). There are weird sentences, as well as many definite articles in some sentences. YAML is not specified.

Shortened journal names should be used in the references. Besides this fact, they are well-formatted.
